# Transient Disappearance of RAS Mutant Clones in Plasma: A Counterintuitive Clinical Use of EGFR Inhibitors in RAS Mutant Metastatic Colorectal Cancer

**DOI:** 10.3390/cancers11010042

**Published:** 2019-01-04

**Authors:** Cristina Raimondi, Chiara Nicolazzo, Francesca Belardinilli, Flavia Loreni, Angela Gradilone, Yasaman Mahdavian, Alain Gelibter, Giuseppe Giannini, Enrico Cortesi, Paola Gazzaniga

**Affiliations:** 1Department of Radiological, Oncological and Pathological Sciences, Sapienza University of Rome, V.le Regina Elena 324, 00161 Rome, Italy; cristina.raimondi@uniroma1.it (C.R.); agelibter@yahoo.it (A.G.); enrico.cortesi@uniroma1.it (E.C.); 2Department of Molecular Medicine, Sapienza University of Rome, V.le Regina Elena 324, 00161 Rome, Italy; chiara.nicolazzo@uniroma1.it (C.N.); francesca.belardinilli@uniroma1.it (F.B.); flavia.loreni@uniroma1.it (F.L.); angela.gradilone@uniroma1.it (A.G.); yasaman.mahdavian@uniroma1.it (Y.M.); giuseppe.giannini@uniroma1.it (G.G.); 3Institut Pasteur—Cenci Bolognetti Foundation, V.le Regina Elena 291, 00161 Rome, Italy

**Keywords:** metastatic colorectal cancer, circulating tumor DNA, RAS, EGFR inhibitors

## Abstract

Genomic studies performed through liquid biopsies widely elucidated the evolutionary trajectory of RAS mutant clones under the selective pressure of EGFR inhibitors in patients with wild type RAS primary colorectal tumors. Similarly, the disappearance of RAS mutant clones in plasma has been more recently reported in some patients with primary RAS mutant cancers, supporting for the first time an unexpected negative selection of RAS mutations during the clonal evolution of mCRC. To date, the extent of conversion to RAS wild type disease at the time of progression has not been clarified yet. As a proof of concept, we prospectively enrolled mCRC patients progressing under anti-VEGF based treatments. Idylla™ system was used to screen RAS mutations in plasma and the wild type status of RAS was further confirmed through IT-PGM (Ion Torrent Personal Genome Machine) sequencing. RAS was found mutant in 55% of cases, retaining the same plasma mutation as in the primary tumor at diagnosis, while it was found wild-type in 45%. Four patients testing negative for RAS mutations in plasma at the time of progression of disease (PD) were considered eligible for treatment with EGFR inhibitors and treated accordingly, achieving a clinical benefit. We here propose a hypothetical algorithm that accounts for the transient disappearance of RAS mutant clones over time, which might extend the continuum of care of mutant RAS colorectal cancer patients through the delivery of a further line of therapy.

## 1. Introduction

In metastatic colorectal cancer (mCRC), RAS mutational status guides the therapeutic use of EGFR inhibitors; therefore, genotyping cancer tissue is mandatory in routine practice to personalize treatments. Since activating mutations in KRAS results in the constitutive activation of downstream signaling pathways, they confer resistance to inhibition of cell surface receptor tyrosine kinases, including EGFR. Consequently, mutations of the KRAS oncogene is a powerful negative predictive biomarker to identify patients with mCRC who do not benefit from EGFR therapy, and much evidence has been provided that clinical benefit from treatment with EGFR-I is limited to patients with tumors harboring the wild-type KRAS gene. In RAS mutant colorectal cancer, the monoclonal antibody bevacizumab is used with cytotoxic doublets FOLFOX/CAPOX/FOLFIRI in the first-line treatment setting and usually continued in combination with any cytotoxic agent or any combination of cytotoxic agents until disease progression or unacceptable toxicity. In second line setting, patients should be considered for treatment with Bevacizumab post-continuation strategy or Aflibercept or ramucirumab with a change in chemotherapy backbone, while regorafenib or TAS-102 are used as third line options [1]. 

Evidence has been provided that the analysis of circulating tumor DNA (ctDNA) in blood samples is a remarkable surrogate of tumor biopsy for mutations detection, with the unique advantage to allow the monitoring of temporal heterogeneity of cancer during the course of targeted therapies [2,3]. Genomic studies performed through liquid biopsies in mCRC demonstrated that RAS mutant clones might rise in plasma of patients before the onset of secondary resistance to anti-EGFR therapy, generating new hypotheses for blood-guided adaptive therapeutic strategies [4]. Similarly, the disappearance of RAS mutant clones in plasma has been more recently reported, supporting for the first time an unexpected negative selection of RAS mutations during the clonal evolution of mCRC. As proof-of-concept, one patient with a primary RAS mutant mCRC, who switched to wild type- RAS status in plasma after failure of first-line therapy, received second-line treatment with anti-EGFR, achieving a durable clinical benefit [5]. This preliminary observation raises the question of whether liquid biopsy testing might expand the population of anti-EGFR-eligible patients by including those with primary RAS-mutant mCRC who convert to wild type in plasma at the time of disease progression (PD). In the present study, we used ctDNA analysis to monitor RAS clonal evolution in a small population of patients with RAS mutant mCRC, as assessed by tumor biomarker testing of primary tumor tissue. Aims of the study were: (1) to investigate the extent of conversion to RAS wild type disease in plasma at the time of progression; and (2) to explore whether the conversion to a wild type RAS status in plasma might be exploitable for therapeutic purpose, thus expanding the population of anti-EGFR-eligible patients. 

## 2. Results

Eleven mCRC patients (pts) with primary diagnosis of RAS mutant mCRC were prospectively enrolled for ctDNA analysis at the time of disease progression at failure of anti-VEGF based treatments (first line 5 pts; second line 3 pts; third line 3 pts). RAS mutational status, as assessed in plasma samples by Idylla™ system [6], was found mutant in 6 pts (55%), retaining the same plasma mutation as in the primary tumor at diagnosis, and wild-type in 5/11 (45%) (Table 1). Plasma DNA samples for further IT-PGM sequencing were available from 4/5 patients testing negative for RAS mutational status at Idylla™ determination. Neither RAS mutations originally identified in the primary tumor nor de novo RAS mutations were detected in these samples, even searching for very low frequencies alleles (Table 2). The four patients testing negative for RAS mutations in plasma at both Idylla™ system and It-PGM sequencing were considered eligible for treatment with EGFR inhibitors. They were all treated with anti-EGFR based therapies, with concomitant longitudinal monitoring of plasma ctDNA RAS mutational analysis every 2–3 months. Patient 1 had a primary diagnosis of RAS mutant metastatic rectal cancer (NRAS A146T) and his clinical history has been partially published [5]. After failure of first line FOLFOXIRI/BEV therapy, he received a second line with FOLFIRI/CET, according to the absence of any detectable RAS mutation as assessed by plasma ctDNA analysis. The treatment was well tolerated and the patient achieved a partial response which lasted 6.5 months. At December 2017, a mild progression of disease was diagnosed, with pericardial effusion. A second ctDNA analysis was performed on both plasma and pericardial fluid and confirmed in both specimens the absence of any detectable RAS mutations. Given that the patient still had a good performance status and was willing to undergo further therapy, the reintroduction of oxaliplatin, in combination to FOLFIRI/CET, was decided. The patient had a stable disease and symptoms control until last clinical evaluation (August 2018). A third ctDNA analysis (May 2018) confirmed the absence of any detectable RAS mutations. The duration of response to cetuximab-based treatment (second line) is currently of 12 months (Figure 1). Patient 2 had a primary diagnosis of stage III RAS mutant colon cancer (KRAS G12C), with evidence of liver metastasis six months after the diagnosis. She was enrolled at failure of a first line therapy with FOLFOXIRI/BEV. Based on the absence of any detectable RAS mutations in plasma ctDNA, from October to December 2017, she received a second line with FOLFIRI/CET. In January 2018, a CT scan showed stable disease and, according to the confirmed absence of any detectable RAS mutations in plasma, a maintenance therapy with DE GRAMONT/CET was started. Last CT scan (June 2018) still confirmed the stability of disease. Given the absence of any RAS mutations in plasma and considering patient’s preferences, a monotherapy with Cetuximab 500 mg/mq g1 q15 was started. The duration of response to cetuximab-based treatment (second line) is currently of 10 months (Figure 1). Of note, a histopathological revision of the primary tumor tissue was performed in this patient at June 2018, unexpectedly revealing RAS wt status. Patient 3 had a primary diagnosis of mutant RAS mCRC (KRAS G12C). He was enrolled at failure of first line (FOLFIRI-BEV) in August 2017. Based on the absence of any detectable RAS mutations in plasma ctDNA, from September 2017 to April 2018, he received a second line with FOLFIRI/PANITUMUMAB. In February 2018, the plasma analysis was repeated revealing the re-appearance of a G12C mutation. The treatment was continued until clinical evidence of PD (April 2018). At that time, the plasma analysis revealed the appearance of a new NRASG13/V mutation. This patient is currently receiving FOLFOX/BEV. The duration of response to panitumumab-based treatment was six months (Figure 1). Patient 4 had a primary diagnosis of mutant RAS (KRAS G12C) metastatic rectal cancer. She was enrolled at failure of a third line treatment (IRINOTECAN/BEV). Based on the absence of any detectable mutations in plasma ctDNA, from January to March 2018, she received a fourth line with IRINOTECAN/CET. After two months of therapy, serum levels of CEA and CA 19.9 significantly decreased. In April 2018, the patient experienced cognitive symptoms and a CT scan showed brain PD. ctDNA analysis was not repeated due to the clinical deterioration of the patient, who died in May 2018. The duration of response to cetuximab-based treatment in this patient was four months (Figure 1).

## 3. Discussion

The KRAS oncogene cycles between a guanosine diphosphate (GDP) bound (“off” state) to guanosine triphosphate (GTP) bound (“on” state) in response to receptor activation. KRAS activating mutations abolish this intrinsic GTPase activity, resulting in constitutively active K-ras proteins that activate downstream signaling pathways, leading to intrinsic resistance to EGFR inhibitors. Intratumor heterogeneity and drug-selected clonal evolution can account for the pulsatile levels of RAS-mutant clones and for the dynamic and reversible changes of molecular cancer drivers. The pattern of clonal evolution of RAS mutations in cancer is highly heterogeneous, and the evolutionary pressure imposed by treatments can result in positive as well as negative selection of RAS mutant clones at relapse. This mixed pattern of positive and negative selection of RAS mutations observed at progression of disease, previously described in hematological malignancies [7,8], supports the concept that RAS mutations are frequently subclonal also in colorectal cancer [9]. In solid tumors, being metastatic sites often difficult to be re-biopsed in the clinical practice, genomic profiling from liquid biopsies, specifically circulating tumor DNA, offers potential advantages when compared with tumor tissue-based sequencing. In our series of RAS mutant colorectal cancer specifically, the negative selection of RAS mutant clones in plasma might have crucial therapeutic implications. Current guidelines indicate that patients with RAS mutant mCRC should not be considered eligible for an anti-EGFR therapy. In this group of patients, the only biological agents that have demonstrated efficacy with chemotherapy are anti-angiogenic drugs, including bevacizumab, aflibercept and ramucirumab, often with limited expectations of clinical benefit after first line [10]. The case series that we here report demonstrates for the first time that, even though clinically significant RAS mutations are found at the time of diagnosis, the biology of metastatic colorectal cancer still evolves over the course of anti-VEGF containing treatments. The absence of detectable RAS mutations in plasma at the time of progression of disease indicates that the molecular identity of primary tumors should be redefined at each progression and that the treatment strategy for RAS mutant mCRC at diagnosis should not necessarily rely only on angiogenesis inhibition along the entire clinical history of the disease. The absence of detectable RAS mutations in plasma cannot certainly exclude that a RAS mutation might be present in the sample below the assay limit of detection (LOD) (1–5%). The same bias is reported for tissue biopsy analysis where the percentage of mutated alleles should be within the analytical sensitivity range according to the method used. Nevertheless, we believe that this is not noteworthy from a mere clinical point of view since it has been demonstrated that patients with low RAS mutant fraction (0.1–5%) might still benefit from the addition of cetuximab to chemotherapy [11,12,13]. Furthermore, the absence of any detectable RAS mutation in plasma might also be due to the insufficient amount of ctDNA in the sample. Again, the selection of the most optimal cell block, selection of the area for DNA extraction and neoplastic cell content determination are critical steps in the tissue molecular testing as well, possibly leading to false negative results [14]. We here report about the clinical benefit achieved by a small group of RAS mutant primary mCRC patients, treated with chemotherapy plus EGFR inhibitors, according to the absence of any RAS mutations in plasma (using both NGS- and PCR-based methods), after failure of treatment with anti-VEGF. In these patients, the introduction of anti-EGFR agents, which would have not been supported by international guidelines (out of this study), allowed achieving a 12-, 10- and 6-month PFS in the three patients treated in second-line setting and a PFS of four months in the patient treated in fourth-line. The suggested mechanism for the pulsatile levels of mutant and wild-type RAS clones throughout the clinical history of mCRC is the constant presence of molecularly distinct subclones persisting in the tumors, which continuously compete for space and resources, under the stresses applied by tumor microenvironment and therapies. In this scenario, the implementation of platforms to longitudinally monitor clonal evolution in blood promises important clinical implications. Some patients may gain additional clinical benefit from the re-use of cetuximab after having progressed on regimens including the same drug in an earlier treatment line [15]. At the same time, the disappearance of RAS mutant clones at failure of anti-VEGF based therapies in patients with primary mutant mCRC could make them candidates for an anti-EGFR therapy. In line with our data, RAS mutational status of mCRC should be re-defined at each treatment failure and the therapeutic strategy should be re-addressed accordingly, as proposed in the treatment algorithm illustrated in Figure 2. When RAS mutations are found at diagnosis in primary/metastatic tumor tissue, this algorithm suggests re-testing the mutational status of RAS genes in plasma at each progression of disease. In this scenario, the therapeutic choice is longitudinally re-adjusted depending on whether liquid biopsy reveals the negative selection of RAS-mutant clones at progression after any lines of therapy. This allows incorporating an additional treatment option (anti-EGFR plus chemotherapy) in the “continuum of care” aimed at improving survival and quality of life of patients with RAS-mutated mCRC.

## 4. Patients and Methods

### 4.1. Patients

Eleven patients with histologically confirmed primary metastatic colorectal cancer who were molecularly diagnosed as having RAS mutant status on primary tumor were prospectively enrolled at the time of PD from any line of therapy. Inclusion criteria were: histopathologically proven mCRC, mutant RAS status on primary tumor biopsy, progression of disease (RECIST criteria), indication for a further line of treatment, ECOG performance status (PS) 0–2, and adequate organ function. Blood samples were collected from each patient after obtaining informed consent. Permission to perform ctDNA analysis from plasma was obtained from the Regional Ethical Committee (No.:179/16) and the study was conducted in accordance with the Declaration of Helsinki. 

### 4.2. Plasma ctDNA RAS Mutation Analysis 

Blood samples (6 mL) were centrifuged at 1500 rpm for 10 min, and then plasma was removed and further centrifuged at 13,000 rpm for 1 min. Plasma samples were aliquoted and used immediately for ctDNA mutational analysis or stored at −80 °C. Idylla™ (Biocartis, Mechelen, Belgium), a fully automated, real-time PCR based molecular diagnostics system, was used to screen RAS mutational profile from plasma. Specifically, Idylla™ ctKRAS Mutation Assay and Idylla™ ctNRAS/BRAF/EGFR Mutation Assays were used. The first allows the detection of 21 mutations in codons 12, 13, 59, 61, 117, and 146 of the KRAS gene, while the second allows the detection of 18 mutations in codons 12, 13, 59, 61, 117, 146 of the NRAS gene and 5 mutations in codon 600 of the BRAF gene from 1 mL of plasma. Both tests require less than 1 min hands-on time, and have a turnaround time of approximately 120 min each. For every valid PCR curve, Idylla™ calculates a quantification cycle (Cq) value. If the difference between the measured Cq for a RAS mutant PCR signal and the RAS wild-type Cq value, i.e., the ΔCq value, is within a validated range, the sample is characterized as RAS mutation positive, and the specific mutation or mutation group is indicated. Samples having a valid RAS wild-type signal but a ΔCq value outside the validated range are reported as being RAS mutation negative. Results expressed as invalid are due to the presence of inhibitors in the sample, insufficient amplifiable DNA present in the sample, incorrect placement of the sample in the cartridge, or a sample volume that is out of range. As indicated in the manufacturer’s instructions, a Cq value of the KRAS control >25.5 and NRAS control >35.5 indicates that a low amount of cell-free DNA is present in the sample. In such cases, low-frequency mutations may not be detected [6]. Plasma samples showing wild-type RAS status were further processed for IT-PGM sequencing. To this purpose, cDNA was purified from plasma for each patient (1 mL) by Maxwell 16 system (Promega) using Maxwell 16 blood DNA purification kit according to the manufacturer’s instructions. DNA samples were analyzed using the Ion AmpliSeq Colon and Lung Cancer Research Panel V2 (CLV2) containing a single primer pool to amplify hotspots and targeted regions of 22 cancer genes frequently mutated in CRCs (https://www.ampliseq.com): FGFR2, ALK, FGFR3, NOTCH1, MAP2K1, FGFR1, DDR2, ERBB4, ERBB2, AKT1, CTNNB1, EGFR, STK11, MET, PTEN, SMAD4, BRAF, FBXW7, PIK3CA, RAS, and TP53 (Thermo Fisher Scientific, Guilford, CT 06437, USA). Barcoded and adaptor-ligated libraries were constructed using the Ion AmpliSeq Library Kit 2.0 (Thermo Fisher Scientific, Van Allen Way, Carlsbad, CA 92008, USA). Templated spheres were prepared using 100 pM of each library on the Ion One Touch 2.0 machine. Template-positive spheres were loaded into an Ion chip 314 and sequenced by IT-PGM machine (Thermo Fisher Scientific, Van Allen Way, Carlsbad, CA 92008, USA). Raw read mapping was performed by the Torrent Suite Software version 5.2 (Thermo Fisher Scientific, https://github.com/iontorrent/TS). For data analysis, adapter trimming and alignment QC we used the Ion Reporter software 5.10, with parameters previously optimized for liquid biopsy (Thermo Fisher Scientific, https://ionreporter.thermofisher.com/ir/). Further filtering was according to the following parameters: coverage, >1000; allelic imbalance between strands, <0.5; and Phreed quality score, ≥100.

## 5. Conclusions

We here for the first time propose a hypothetical algorithm that accounts for the transient disappearance of RAS mutant clones over time, extending the continuum of care through the delivery of as many lines of therapy as before moving into palliative care (Figure 2). Owing to these preliminary data, we designed a phase II prospective study with the aim to investigate whether targeting the plasma RAS-wild type window with EGFR-inhibitors might represent on a large scale an exploitable second-line option in RAS-mutant mCRC.

## Figures and Tables

**Figure 1 cancers-11-00042-f001:**
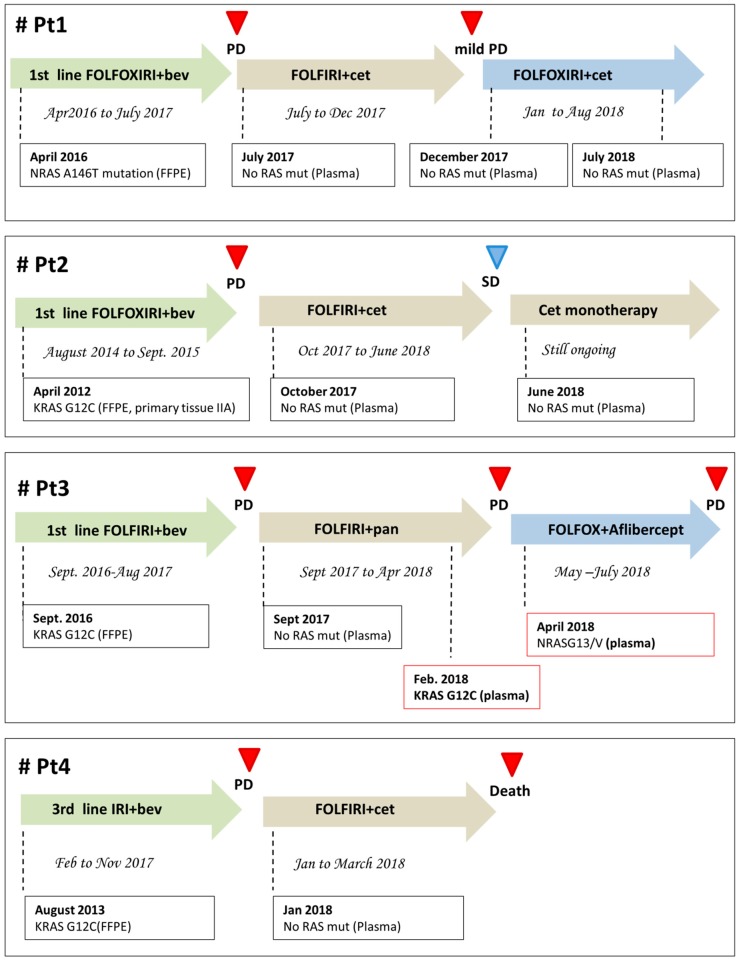
Clinical history of the four patients described in the manuscript. All patients were serially monitored with liquid biopsy for RAS-genes mutations and therapy was re-adjusted accordingly. Bev, bevacizumab; Cet, cetuximab; PD, progression of disease; SD, stability of disease.

**Figure 2 cancers-11-00042-f002:**
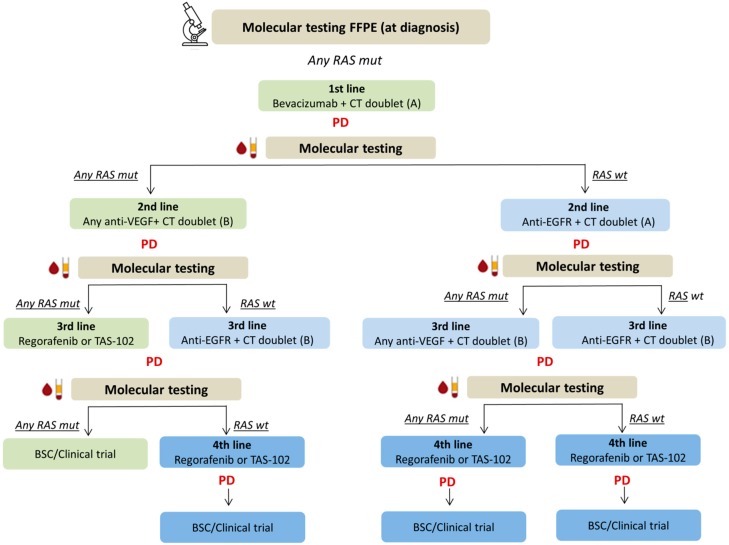
Proposed therapeutic algorithm for RAS-mutated mCRC based on serial molecular testing of RAS genes mutations in plasma. When RAS mutations are found at diagnosis in primary/metastatic tumor tissue, this algorithm suggests re-testing the mutational status of RAS genes in plasma at each progression of disease. In this scenario, the therapeutic choice is longitudinally re-adjusted depending on whether liquid biopsy reveals the negative selection of RAS-mutant clones at progression after any lines of therapy. This allows incorporating an additional treatment option (anti-EGFR plus chemotherapy) in the “continuum of care” aimed at improving survival and quality of life of patients with RAS-mutated mCRC. Standard treatment options for patients who maintain RAS mutations in blood are indicated in green. Therapeutic options for patients who convert to RAS-wild type at any point are indicated in blue (additional line of therapy as compared to standard is highlighted in dark blue). BSC, best supportive care; CT, chemotherapy; FFPE, formalin-fixed paraffin-embedded; mCRC, metastatic colorectal cancer; PD, progression of disease.

**Table 1 cancers-11-00042-t001:** Demographic of patients.

All Patients (*n* = 11)	
**Age**	
Median age (years)	62.2
Range (years)	49 to 78
**Sex**	
Female	3
Male	8
**Site of Metastasis**	
Liver	10
Peritoneum	1
**RAS Mutation FFPE**	
KRAS G12C	3
KRAS G12V	3
KRAS G12D	1
KRAS G12A	1
KRAS G13D	1
KRAS Q61R/L	1
NRAS A146T	1
**Previous Lines of Therapy** **(at the Moment of Liquid Biopsy)**	
One	8
Three	3
**RAS Mutation Plasma** **(at the Time of Radiological PD)**	
Wild-type	5
KRAS G12D	2
KRAS G12A	1
KRAS G12V	1
KRAS G13D	1
KRAS Q61R/L	1

**Table 2 cancers-11-00042-t002:** IT-PGM sequencing on DNA extracted from plasma.

Sample	*RAS* Somatic Mutations (VAF)	*Other Genes* Somatic Mutations (VAF)	*Other Genes* Germline Mutations (VAF)
1	WT		*ERBB4* c.421 + 58A > G (45.2%) *FGFR3* c.1953 G > A p.Thr651 = (99.9%) *TP53* c.639T > G p.Arg213 = (59.4) c.215C > G p.Pro72Arg (98.4%)
2	WT		*FGFR3* c.1953 G > A p.Thr651 = (99.8%) *EGFR* c.1498 + 22 (99.2%) c.2361G > A p.Gln787 = (51.7%) *MET* c.534C > T p.Ser178 = (49.4%) *TP53* c.215C > G p.Pro72Arg (98.5%)
3	WT	*FGFR1* c.443G > A p.Arg148His (2.31%) *TP53* c.659A > G p.Tyr220Cys (1.01%)	*ERBB4* c.421 + 58A > G (51.2%) *FGFR3* c.1953 G > A p.Thr651 = (99.0%) *EGFR* c.1498 + 22A > T (99.9%) c.2361G > A p.Gln787 = (99.4%) *TP53* c.215C > G p.Pro72Arg (45.3%)
4	WT		*FGFR3* c.1953 G > A p.Thr651 = (99.8%) *EGFR* c.1498 + 22 (99.9%) c.2361G > A p.Gln787 = (99.8%)

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
