# Peer review of "Transient Disappearance of RAS Mutant Clones in Plasma: A Counterintuitive Clinical Use of EGFR Inhibitors in RAS Mutant Metastatic Colorectal Cancer"

_cancers, 2019, doi:10.3390/cancers11010042_

Reviewer 1 Report

The authors explore whether the conversion to a wild type RAS status in plasma might be exploitable for  a therapeutic purpose in metastatic colorectal cancer, in order to expand the population of anti-EGFR-eligible patients. This referee considers the topic of the manuscript is of high interest to oncologists and to the scientific community since it provides clinical data supporting a novel treatment algorithm for metastatic colorectal cancer at the time of disease progression, which constitutes a major issue in the clinic.

This referee believes that some modifications should be done to the manuscript in order to clarify some issues.

Here follows the list of concerns:

1-    In “Introduction” section (page 2), the authors indicate that RAS mutational status guides the therapeutic use of EGFR inhibitors in metastatic colorectal cancer (mCRC). In results section they mentioned the different treatment regimens given to the patients. This reviewer believes that the article would benefit from a description of the different lines of treatment for mCRC in the introduction section, so that when in the “Results” section the authors describe the treatments corresponding to each patient the reader would better understand the treatment regimens. Besides, it would also be interesting to include a description of the different EGFR inhibitors (such as cetuximab or panitumumab) as well as their mechanisms of action.

2-    In “Result” section, in the paragraph describing the treatment regimens corresponding to Patient 2, this reviewer believes that an explanation of the reasons for changing the therapy from FOLFIRI/CET to DE GRAMONT/CET should be included within the text.

3-    In “Discussion” section (page 6, line 125) the authors mentioned: “Current guidelines indicate that patients with RAS mutant mCRC should not be considered eligible for an anti-EGFR therapy”. This reviewer believes that the manuscript would benefit from the inclusion of a paragraph providing the mechanistic explanation for which RAS mutant mCRC becomes resistant to anti-EGFR therapy.

4-    Patient #2 had a primary diagnosis of stage III RAS mutant colon cancer (KRAS G12C). However, an histopathological revision of the primary tumor tissue unexpectedly revealed RAS wt status. The authors discuss the fact that the actual techniques have false negative or positive results. This is really important because these false results guide to the wrong treatment regimen. This reviewer would like to know if there are statistics regarding the percentage of false negative or positive results corresponding to RAS mutational status and how this affect the outcome of the patients. This reviewer believes that this information should be included in the “Discussion” section.

5-    Finally, the authors propose an hypothetical therapeutic algorithm for RAS-mutated mCRC based on serial molecular testing of RAS genes mutations in plasma. This reviewer believes that this algorithm should be described in detail in the “Discussion” section.

Some minor comments.

-       Please indicate the meaning of the following abbreviations within the text:

Page 1, line 26: IT-PGM sequencing.

Page 1, line 29: PD (disease progression)

Page 2, line 61: pts (patients)

Page 6, line 137: LOD

Page 7, line 189: IFU

Please note that the reference is missing in page 2 line 72.

Author Response

Reviewer 1:

we thank for the useful suggestions.

Concern1: we included all the information required in the introduction and discussion section

Concern 2: the patient underwent a maintainance therapy with de gramont as suggested by international guidelines

Concern 3: we included  a phrase on the explanation for which RAS mutant mCRC becomes resistant to anti-EGFR therapy. Actually, it is a very known mechanism and it is not an acquired resistance, but an intrinsic mechanism.

Concern 4: There are no statistics regarding the percentage of false negative or positive results corresponding to RAS mutational status.  What is largely known (and already discussed in the discussion section) is that different pathologists work with different methods to detect RAS mutations. So it’s not a concern of “false positive”, but a concern of allelic frequency. Pathologists working with highly sensitive methods risk to give the result “mutant” to all cases with an allelic frequency between 1 and 5%. But we know from clinical trials that tumors with low allelic frequencies still respond to EGFR inhibitors.

Concern 5: we included a full explanation of the algorithm in the discussion section

All minor concerns have been addressed and the text has been corrected accordingly

Reviewer 2 Report

In this manuscript the authors investigated RAS mutations in ctDNA of eleven mCRC patients with primary diagnosed RAS-mutations after disease progression. In four of these patients no RAS mutation could be detected in ctDNA at study inclusion. Three of them still show no ctDNA RAS mutation at later time points. All patients were treated with cetuximab or panitunumab. One patients showed a mild PD, one a stable disease, one PD and one died with anti-EGFR treatment. The authors suggest to re-define RAS status in mCRC patients at each treatment failure to adapt therapeutic strategy.

RAS assessment is mandatory for therapy decision in mCRC patients especially if anti-EGFR therapy is considered. Several papers have demonstrated that RAS status can be determined using ctDNA with an overall agreement to the primary tumor of more than 90%. However it has also been shown that plasma/tissue RAS discrepancies can be explained by spatial and temporal tumor heterogeneity. Analysis of clinico-pathological features showed that the site of metastasis (i.e. peritoneal, lung), the histology of the tumor (i.e. mucinous) and administration of treatment previous to blood collection negatively impacted the detection of RAS in ctDNA. These parameters have not be considered in this paper. Therefore, it is difficult to estimate if the patients really lost RAS mut due to clonal evolution of if it is a detection problem. The low patient number and the heterogeneity in outcome makes it difficult to estimate the benefit of an anti-EGFR therapy in these patients. Furthermore, the authors did not describe how other detected gene mutations may contribute to the outcome of these patients (e.g. EGFR mutations). A larger cohort is needed to state that adaptation of therapy after monitoring of RAS mutations during course of disease is useful as it is planned by the authors.

Author Response

Reviewer 2:

We agree with the reviewer that the site of metastasis (i.e. peritoneal, lung), the histology of the tumor (i.e. mucinous) and administration of treatment previous to blood collection negatively impact the detection of RAS in ctDNA. Anyway, we used two different methods (NGS and PCR based) to confirm the wt status of RAS. The presence of other somatic mutations in blood identical to those observed in primary tumor (see pt. 3) clearly confirms that ras mutation can be lost during disease progression.  We also agree that the low patient number and the heterogeneity in outcome makes it difficult to estimate the benefit of an anti-EGFR therapy in these patients. This is the reason why we decided to submit as “concept paper”. A clinical trial is planned for 2019 and it is under our CE approval. It will also answer the question  whether other detected gene mutations may contribute to the outcome of these patients